# Trends of Studies on Controlled Halogenated Gases under International Conventions during 1999–2018 Using Bibliometric Analysis: A Global Perspective

**Jing Wang** [1,†], **Hui-Zhen Fu** [2,†], **Jiaqi Xu** [1], **Danqi Wu** [1]⬤, **Yue Yang** [1], **Xiaoyu Zhu** [1] **and Jing Wu** [1,*]

1 The MOE Key Laboratory of Resource and Environmental System Optimization, College of Environmental Science and Engineering, North China Electric Power University, Beijing 102206, China; 120192232390@ncepu.edu.cn (J.W.); jx2466@columbia.edu (J.X.); wudanqi8592@163.com (D.W.); e0732769@u.nus.edu (Y.Y.); zhuxiaoyu@ncepu.edu.cn (X.Z.)

2 Department of Information Resources Management, School of Public Affairs, Zhejiang University, Hangzhou 310058, China; fuhuizhen@zju.edu.cn

* Correspondence: wujing.108@163.com; Tel.: +86-1061772891

† Jing Wang and Hui-Zhen Fu contributed equally to the article.

**Abstract:** A lot of research on international convention-controlled halogenated gases (CHGs) has been carried out. However, few bibliometric analyses and literature reviews exist in this field. Based on 734 articles extracted from the Science Citation Index (SCI) Expanded database of the Web of Science, we provided the visualisation for the performance of contributors and trends in research content by using VOSviewer and Science of Science (Sci2). The results showed that the United States was the most productive country, followed by the United Kingdom and China. The National Oceanic and Atmospheric Administration had the largest number of publications, followed by the Massachusetts Institute of Technology (MIT) and the University of Bristol. In terms of disciplines, environmental science and meteorological and atmospheric science have contributed the most. By using cluster analysis of all keywords, four key research topics of CHGs were identified and reviewed: (1) emissions calculation, (2) physicochemical analysis of halocarbons, (3) evaluation of replacements, and (4) environmental impact. The change in research substances is closely related to the phase-out schedule of the Montreal Protocol. In terms of environmental impact, global warming has always been the most important research hotspot, whereas research on ozone-depleting substances and biological toxicity shows a gradually rising trend.

**Keywords:** global research trend; SCI-Expanded database; scientometrics; halogenated gases; climate change; ozone depletion

## 1. Introduction

Halogenated gases deplete ozone and contribute to global warming and have received widespread attention. One of the major characteristics is their extremely high reactivity with electrons [1]. When they reach the stratosphere after being emitted from the Earth's surface, they absorb ultraviolet radiation and decompose, generating halogen radicals. Halogen radicals are involved in very effective catalytic chain reactions that deplete the ozone layer, causing a decrease in the ozone concentrations of the stratosphere. Ozone depletion allows more solar ultraviolet-B radiation (290–320 nm wavelength) to reach the surface [2], which can cause severe harm to animals, plants, and microorganisms [3–5]. In addition, these halogenated gases are potent greenhouse gases. The Synthesis Report (SYR) of the Intergovernmental Panel on Climate Change (IPCC) Fifth Assessment Report [6] highlighted that the cumulative radiative forcing of all halocarbons from 1750 to 2011 accounted for approximately 13% of the total radiative forcing of greenhouse gases. Due to the dual environmental impact of halogenated gases, the Montreal Protocol and its amendments included chlorofluorocarbons (CFCs), hydrochlorofluorocarbons (HCFCs),

and carbon tetrachloride ($CCl_4$) as ozone-depleting substances that need to be regulated. Subsequently, hydrofluorocarbons (HFCs), perfluorocarbons (PFCs), sulfur hexafluoride ($SF_6$), and nitrogen trifluoride ($NF_3$) were listed by the Kyoto Protocol and the Paris Agreement as greenhouse gases that need to be regulated. The Kigali Amendment to the Montreal Protocol called for the phase-down of HFCs in 2016. Hence, the research object of this study is the above-mentioned international convention controlled halogenated gases (CHGs).

The international community attaches great importance to the impact of ozone depletion on the environment. The United Nations Environment Programme (UNEP) organised the Environmental Impact Assessment Committee for Ozone Depletion. Since 1988, research progress on the impact of ozone layer depletion on the environment has been announced to the world in the form of assessment reports every four years. Since the implementation of the Montreal Protocol, ozone depletion has been alleviated to a certain extent [7,8]. Stratospheric ozone is expected to return to the 1960 levels by the end of the 21st century [9]. However, recent scientific studies have found that some CHG concentration and emission trends are different from those expected. For example, the study of Montzka et al. [10] shows that although reported production has been close to zero since 2006, CFC-11 emissions have increased by $13 \pm 5$ gigagrams ($25 \pm 13\%$) per year since 2012. This discovery brought more attention to CHGs.

The greenhouse effect and global warming caused by CHGs have also received wide international attention, and the focus has increased more in recent years. IPCC has published reports on climate change since 1990. The reports show the growth trend of halogenated compounds over several decades and indicate that the atmospheric content of HFCs, PFCs, $SF_6$, etc. has increased rapidly since the 1990s. In April 2021, at the China-France-Germany Video Summit, China announced that it decided to accept the Kigali Amendment to strengthen the control of non-$CO_2$ greenhouse gases such as HFCs. The joint statement issued by China and the United States in response to the climate crisis also highlighted that the two countries will separately implement measures to gradually reduce the production and consumption of HFCs.

CHGs are concerned with two major scientific issues of global concern; however, there are few quantitative summaries and critical reviews in this field. Therefore, it is necessary to systematically summarise the literature and clarify the research hotspots and future research trends in this field.

Bibliometrics was first introduced by Pritchard (1969). It has been widely used as an effective and useful tool for evaluating scientific results and research topics in specific research fields [11,12]. The performance of national contributors, institutional contributors, and authors at different levels is an important factor in understanding a field [13,14]. These studies assume that the number of publications of a country in a specific scientific subfield reflects its commitment to the state of science and is a reasonable indicator of its contribution to research and development in that field. Collaboration has intensified in recent years owing to the rapid development of scientific communication [15]. Collaboration also leads to a higher citation impact in practically all science areas [16–18]. Therefore, the study of collaboration patterns between researchers and regions could provide important references for other scientific researchers and policy managers.

Another important concern in bibliometric studies is the identification of the research topics. The performance of contributors and their collaborations is not a complete indication of trends or future directions in the research field [19]. Information closer to the study itself, including source title, author keyword, keywords plus, and abstracts [20,21], should be introduced to study the research trend. The analysis combining the words in the title, author keywords, and keywords plus could minimise some limitations, such as the incomplete meaning of single words in the title, small sample size for author keywords, and indirect relationship between keywords plus and the research emphases [11]. These types of words are checked by time periods to show the trends and to minimise year-to-year fluctuations. Therefore, word cluster analysis combining author keywords, keywords plus, and title

content words has proved to be a more effective and comprehensive bibliometric method, which has been successfully applied to reveal research trends and hotspots in the research fields of risk assessment [22], drinking water [14], and pluripotent stem cells [23]. The researchers express the opinion that the collaborative application of co-occurrence analysis and word cluster analysis can shed light not only on research trends, but also on the role of landmark works in the evolution of the research field.

In recent years, there has been increasing interest in visualising scientometrics using data mining and information retrieval to uncover possible collaborative behaviour patterns among contributors. VOSViewer [18] and Science of Science (Sci 2) [24] are newly developed tools that are interactive visualisation and exploration platforms for various networks and complex systems. These tools have been used to develop interactive superposed scientific maps based on the relationship between text words [25], co-author [26], and cooperation among research institutes [27].

Based on the relevant publications retrieved from the web of science database from 1990 to 2018, this study carried out a bibliometric measurement of the CHGs field. Through the quantitative analysis of the literature, the contributions of countries/regions, institutions, individuals, and disciplines are studied. More importantly, through co-citation analysis and word clustering analysis, the themes and hotspots in the field of CHGs are elaborated to provide a quick and in-depth understanding of the field.

## 2. Methodology

### 2.1. Data Collection

Data were obtained from the online version of the Science Citation Index (SCI) Expanded database of the Web of Science from Thomson Reuters in September 2019. ("chlorofluorocarbon*", "hydrochlorofluorocarbon*", "hydrofluorocarbon*", "hydrofluoroolefin*", " perfluorocarbon*", "sulphur hexafluoride", "nitrogen trifluoride", "halons", "carbon tetrachloride", "methyl bromide", "bromochloromethane", "dichloromethane", "chloroform", "trichloromethane", "perchloroethylene") AND ("global warming", "climate warming*", "ozone deplet*", "climate chang*", "climatic chang*", "Greenhouse gas", "radiation forc*") were searched in terms of topic within the publication years of 1999–2018. A total of 1116 publications met the inclusion criteria. Journal articles were selected for further analysis because they are the predominant article type, and the entire research objectives and results are also included in the article [20]. The "front page" was another filter condition [28]; therefore, only articles that contain search terms in the text of their "front page" (including article titles, abstracts, and keywords) were included. This resulted in 734 publications on CHGs over an 18-year period that were considered herein.

To obtain an overview of CHG research, the number of articles published annually from 1999 to 2018 is shown in Figure 1. The number of CHGs publications increased with several fluctuations from 31 in 1999 to 60 in 2018. An increasing number of journals published articles on CHGs. The average article length fluctuated slightly, with an overall average of 8.6 pages. In 1999, there were 40 references per paper, whereas in 2018, there were 48 references per paper—a slight increase over the past two decades.

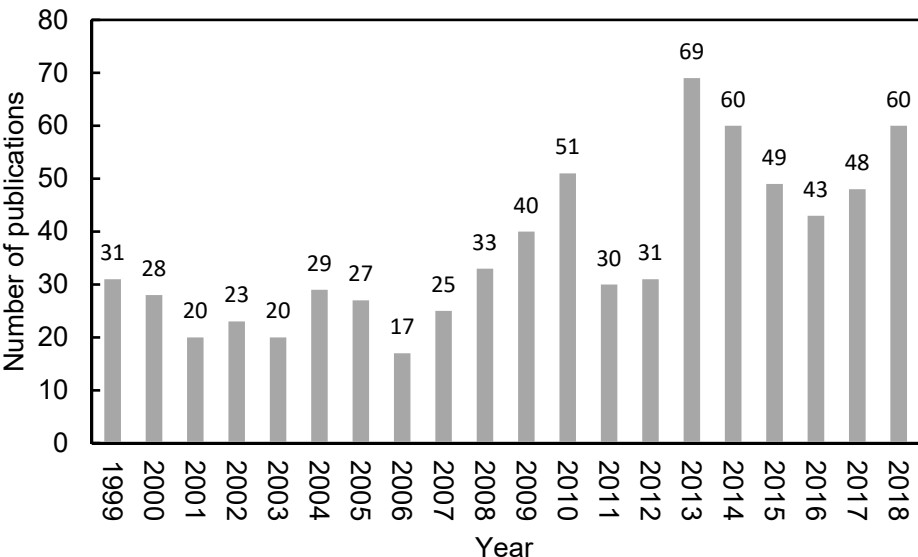

**Figure 1.** Annual number of publications on CHGs research during 1999–2018.

*2.2. Methods*

The downloaded content included author names, title, journal, abstract, contact addresses of the authors, year of publication, keywords, and keywords plus and Web of Science categories of the article. Records were downloaded into a spreadsheet; the country of origin of the collaborator, impact factor of the journals, and number of authors for additional coding [29] were incorporated manually. Articles from England, Scotland, Northern Ireland, and Wales were defined as coming from the United Kingdom (UK). Articles originating from Hong Kong were classified as originating from China. The contributions of different countries and institutions were set to include at least one author in the publication. The type of collaboration was determined by the researchers' address; if all the researchers were from the same country, the term "single-country article" was specified. The term "international collaborative articles" referred to articles written jointly by researchers in more than one country.

VOSviewer was developed by researchers at Leiden University in 2007. It is used to build a visual bibliometric network, such as researcher collaboration networks. VOSviewer also provides text mining functions, creating a visual co-occurrence network of important terms extracted from scientific literature [30]. In this study, VOSviewer was used to analyse the collaboration networks of contributors and co-occurrence networks. The Science of Science (Sci2) Tool is a modular toolset specifically designed to visualise scholarly datasets, study of science, network analysis, and supporting geospatial [24]. In this study, the Sci2 Tool was used to analyse the global geographic distribution of publications and the collaboration model of publications.

## 3. Performance of Contributors

*3.1. Macro Contributors of Country/Territory*

Cooperation between countries was discussed in depth using the Sci2 and Gephi tools. Figure 2 reveals the different patterns of the global geographic distribution of CHGs research publications. The shades of yellow to blue colour correspond to the total number of publications in the country from 1999 to 2018. The deeper the shade, the more papers the country publishes. The lines between any two countries represent a cooperative relationship between them. The thicker the line, the more intensive the international cooperation between the two countries.

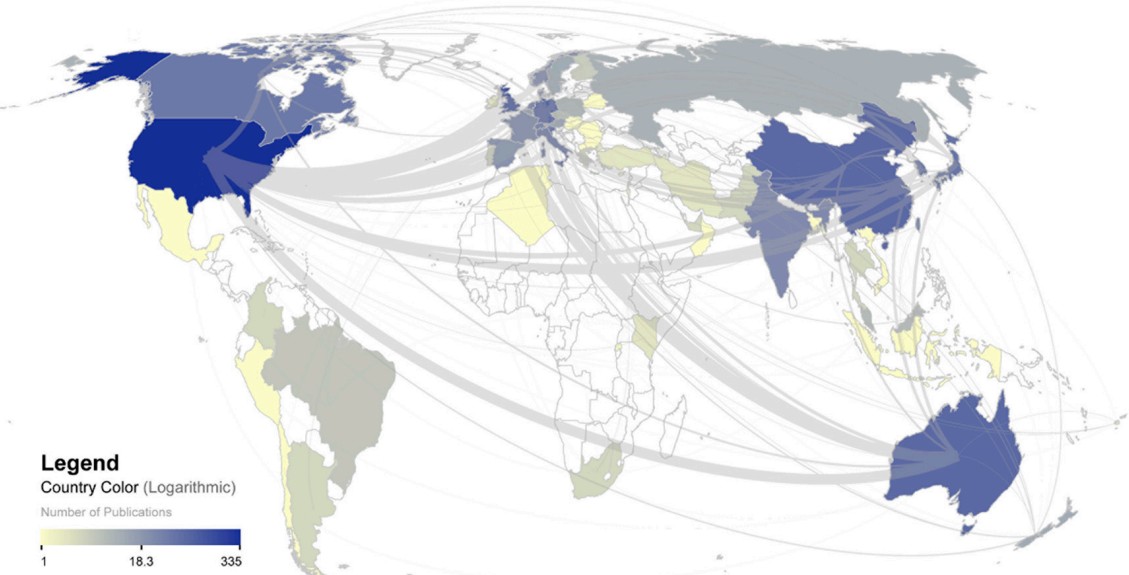

**Figure 2.** Global geographical distribution of publications and collaboration patterns of CHGs publications. Color represents the number of publications between 1999 and 2018; the darker the color, the greater the number of publications. Lines between any two countries represent a cooperative relationship between them. The thicker the line, the more intensive the international cooperation between the two countries.

Judging from the geographical distribution of publications, the number of countries/regions contributing articles to CHGs publications has increased significantly. Therefore, international cooperation has been greatly strengthened. The United States of America (USA) showed the highest contribution with 335 articles (46%), followed by the UK (125 articles; 17%) and Germany (75 articles).

To demonstrate international collaboration, Figure 2 shows the current partnerships among the 60 countries with high citation rates in the field. The map has 60 nodes and 263 undirected weighted edges, indicating 263 cooperative country pairs for the 60 countries. Nodes with more international cooperation articles are larger, whereas countries with more cooperation are connected through thicker edges. The United States is at the centre of the global network of cooperation. The US–UK collaboration was the strongest with 60 articles, followed by the Australia–UK collaboration with 35 articles. The United States is the most favoured national scientific partner in the field of electronics and electrical engineering, possibly due to its high level of research and its leading position. About half of the pairs (129 out of 263) had only one article. Other countries have not developed significant research networks between them, possibly because of the small number of publications. It was not surprising that countries with fewer publications dominate because this pattern has emerged in most scientific fields [31].

It should be noted that there is little research on CHGs in emerging countries. The rapid development and industrialization of those regions meant that large quantities of halogenated gases could be released from the region and cause damage to the ozone layer. It would be interesting to further stimulate more research in those regions and to foster international collaboration with more mature research teams.

### 3.2. Meso Contributors of Institution

The top 10 institutions in terms of productivity are listed in Table 1. Research institutes are concentrated in North America, Europe, and Asia. National Oceanic and Atmospheric Administration (NOAA) (65) published the most articles in North America, the University of Bristol (51) published the most articles in Europe, and the Peking University published the most articles in Asia (14). NOAA had the greatest number of publications with a total

of 65 papers. At the second position is the Massachusetts Institute of Technology (MIT), with 52 publications, followed by the University of Bristol (51 publications), University of Colorado (50 publications), and University of California-San Diego (UCSD, 48 publications). In terms of citations, NOAA (3107 publications) was the most prolific institution, followed by MIT (2577), UCSD, National Aeronautics and Space Administration (NASA, 2489), the University of Bristol (2357), National Center for Atmospheric Research (NCAR) (2123), and Commonwealth Scientific and Industrial Research Organization (CSIRO) (2078).

**Table 1.** Top 10 research institutions in the field of CHGs.

| Rank | The Name of Institution | Number of Publications | Citations/Publications | Number of Collaborators |
|------|-------------------------|------------------------|------------------------|-------------------------|
| 1 | NOAA | 65 | 48 | 63 |
| 2 | MIT | 52 | 50 | 49 |
| 3 | Univ Bristol | 51 | 46 | 48 |
| 4 | Univ Colorado | 50 | 44 | 46 |
| 5 | UCSD | 48 | 52 | 52 |
| 6 | NASA | 47 | 53 | 44 |
| 7 | NCAR | 28 | 76 | 34 |
| 8 | UNIV CALIF IRVINE | 27 | 63 | 36 |
| 9 | CEIRO | 25 | 83 | 38 |
| 10 | Ford Motor Company | 20 | 882 | 10 |

The co-authorship institutional analysis network had a minimum threshold of five publications. The cooperative relationships among 87 institutions are shown in Figure 3. Each node in the figure represents an institution, the size of nodes represents the number of articles, the line between nodes represents the cooperation between institutions, and the thickness of the line represents the link strength between institutions. NOAA and the University of Colorado were the most strongly linked with 32 articles. In addition, the institutions with more cooperation are MIT, UCSD, and Univ Bristol. The cooperation between each of these organizations has reached more than 26. These phenomena are very reasonable. The work of NOAA Earth System Research Laboratories (ESRL) is dominated by its work in University of Colorado Boulder, so it is no surprise that there is a strong NOAA–Univ Colorado connection. MIT, UCSD, and Univ Bristol are research institutions of Advanced Global Atmospheric Gases Experiment (AGAGE), one of the most advanced, most systematic, and most contributing international observation networks for ozone-depleting substances (ODS) and fluorine-containing greenhouse gas observation technologies, and the results are shared among member institutions. This network is mainly sponsored by NASA's Atmospheric Composition Focus Area in Earth Science, and its research institutions also include CSIRO, Swiss Federal Laboratories for Materials Science and Technology (EMPA), University of Urbino, etc. It can be seen in Figure 3 that the AGAGE member units are all marked in blue, which shows that they have close cooperation and similar research.

### 3.3. Micro Contributors of Authors

The minimum number of publications for each author was set to five, and 82 authors were screened. Some of the 82 authors in the network were not connected to each other. To improve the visualisation, we eliminated unconnected authors, and finally presented 39 authors in the final network map of co-authorship authors. Due to this process, some authors do not appear in Figure 4.

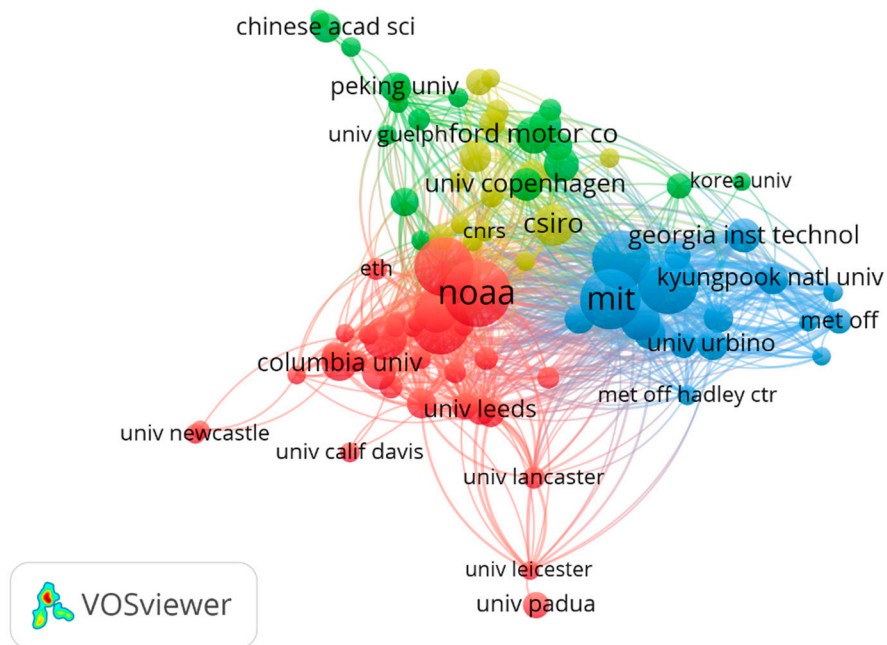

**Figure 3.** Cooperation relationship among different institution. Dots represent institutions, while dot size represents the number of published documents. Lines between dots indicate a connection between two institutions; thicker lines show a stronger connection and indicate a higher number of collaborated articles.

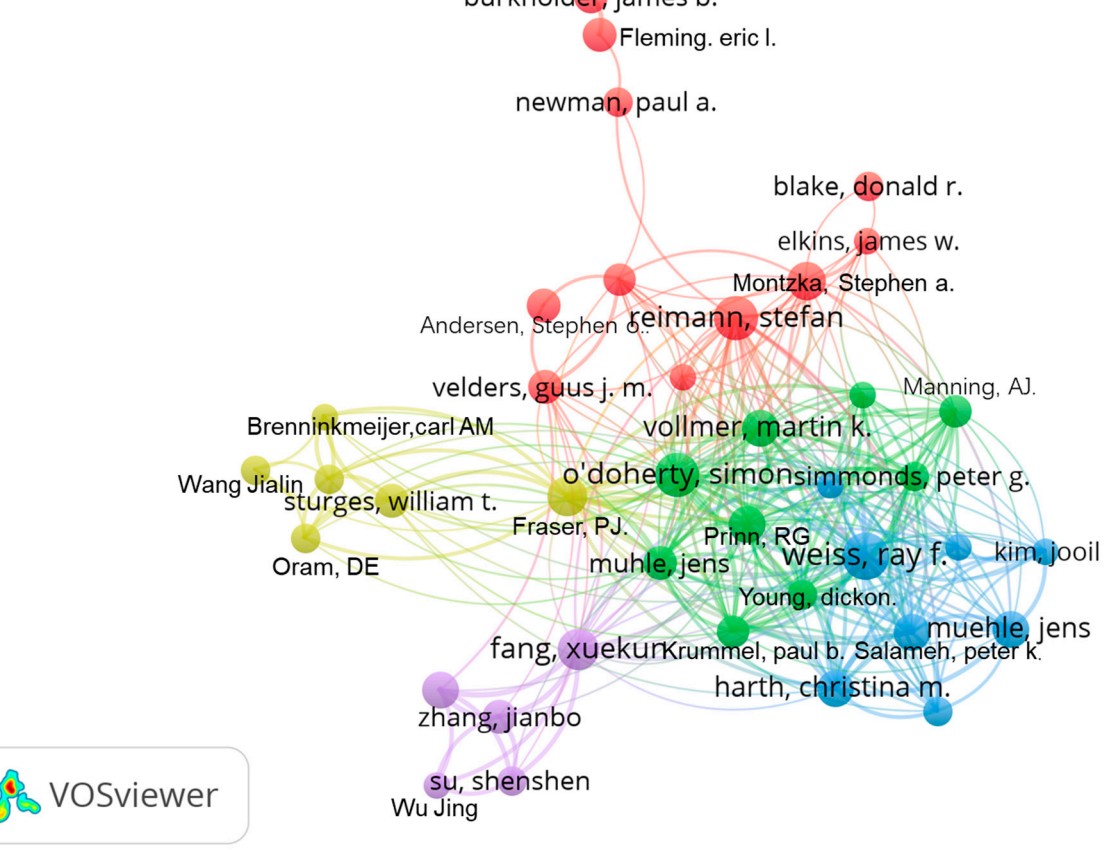

**Figure 4.** Cooperation relationship among different authors. Each node represents an author, the size of nodes represents the number of papers, the line between nodes represents the cooperation between authors, and the thickness of the line represents the link strength between authors.

McCulloch, A was the most prolific author, with 16 publications, followed by Weiss, RF with 15 publications. The top 10 authors are listed in Table 2. There were five clusters with different colours; authors in the same cluster usually suggested that they studied a similar field and had close cooperation with each other. There are two clusters of authors who collaborated more with others: blue cluster represented by Weiss, RF, Salameh, PK, Harth, CM, Jens Mühle, etc. (the main research direction was atmospheric observation); green cluster represented by Rigby, M, O'Doherty, S, and Martin K, whose main research direction was emission estimation. The authors in the red and yellow clusters are also doing observation and emission research, but there are slight differences. The research in yellow clustering is more inclined to aircraft-based observations, and more emphasis is placed on the emission of short-lived halocarbons. The main research direction of the author in purple clustering is to establish emission inventories based on production and consumption data.

**Table 2.** Top 10 authors ranked by publications.

| Rank | The Name of Author | Number of Publications | Citations/Publications | Number of Collaborators |
|------|--------------------|------------------------|------------------------|-------------------------|
| 1 | McCulloch, A | 16 | 102 | - |
| 2 | Weiss, RF | 15 | 23 | 25 |
| 3 | O'Doherty, S | 13 | 8 | 27 |
| 4 | Reimann, S | 13 | 10 | 25 |
| 5 | Fang, Xuekun | 12 | 8 | 23 |
| 6 | Fraser, PJ | 11 | 12 | 28 |
| 7 | Simmonds, PG | 10 | 135 | 21 |
| 8 | Montzka, SA | 10 | 20 | 19 |
| 9 | Shine, KP | 10 | 61 | - |
| 10 | Prinn, PG | 9 | 4 | 27 |

## 4. Research Topics

### 4.1. Macro Topic of Category

To show the interdisciplinarity and distribution of disciplines, the interdisciplinary relationship among 90 disciplines in the current CHGs field was analysed (Figure 5). In total, 90 categories and 189 undirected weighted edges are present on the map, indicating 189 interdisciplinary pairs in these 90 categories. The node size is directly proportional to the number of articles published. Thicker edge connections represent more frequent interdisciplinary categories. Environmental sciences represent the largest macro-field with 650 articles, followed by physical sciences (456 articles), life sciences, and medicine (138 articles). Environmental sciences contributed the most (248 articles, 34%), followed by meteorology and atmospheric sciences (147 articles, 20%), and environmental engineering (71 articles, 9.6%). The interdisciplinarity of the environmental sciences, meteorology, and atmospheric sciences was the strongest with 100 articles, followed by the interdisciplinary pair of environmental sciences and environmental engineering with 64 articles, and the pair of the mechanical engineering and thermodynamics with 30 articles.

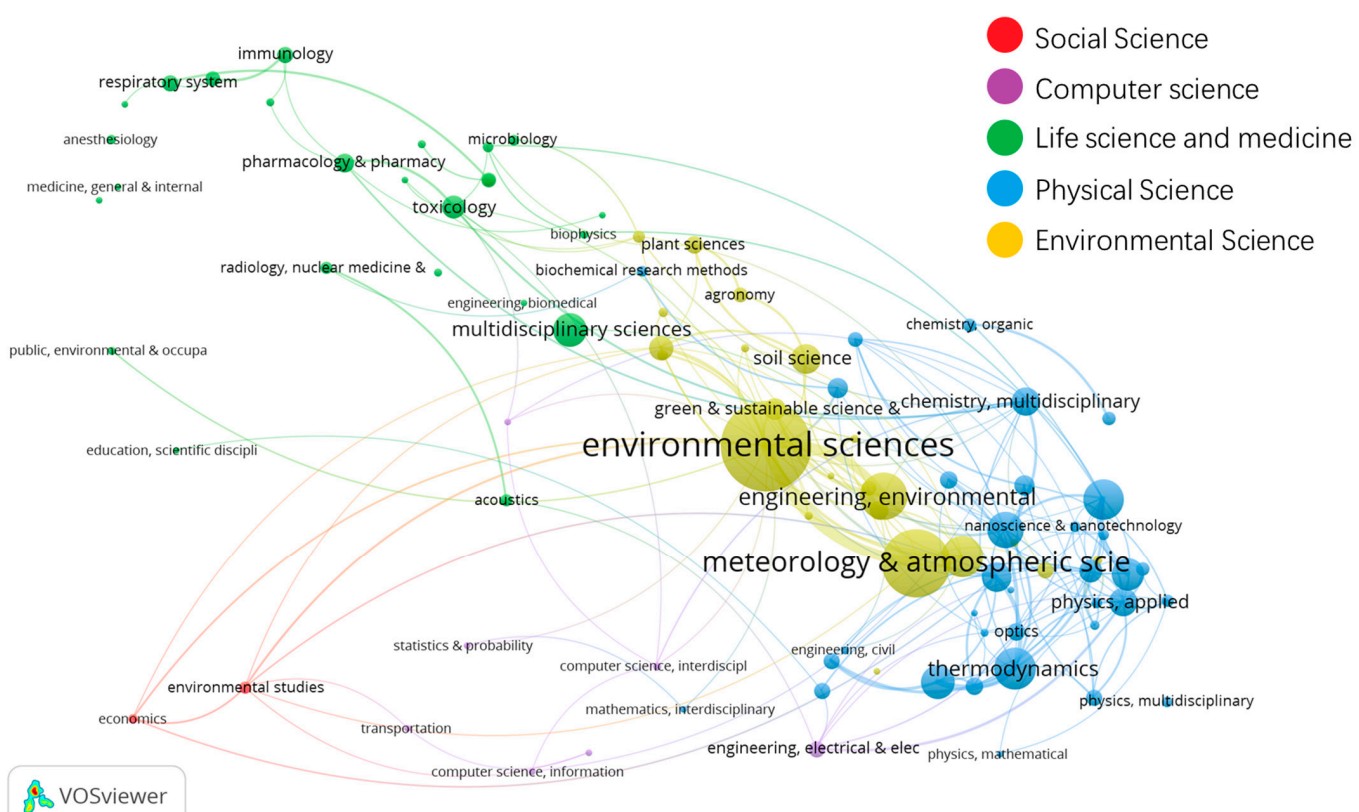

**Figure 5.** Current interdisciplinary relationships among categories of CHGs field. Nodes show web of science categories used to categorize CHGs research articles. The node size is directly proportional to the number of articles published. Thicker edge connections represent more frequent interdisciplinary categories.

### 4.2. Micro-Topic of Keywords

High-frequency keywords can reflect research hotspots. A total of 4526 co-occurrence keywords were extracted from 734 articles. The minimum occurrence of each keyword was set to eight times, and 181 co-occurrence keywords were finally presented. A keyword co-occurrence network map is shown in Figure 6. The top three keywords ranked by number of occurrences were as follows: emission (*n* = 89), chlorofluorocarbons (*n* = 86), and ozone (*n* = 85).

Nodes with the same colour belong to a cluster. The keywords were classified into four clusters. This study sorts out the topics of each cluster by reading the articles contained in each cluster to provide references for future research directions.

4.2.1. Cluster 1 (Green): Research on CHGs Emissions Calculation

Keywords: emission, halocarbons, in situ measurements, mixing ratios, global emissions, European emissions.

Two types of methods, bottom-up and top-down methods, are often used to calculate CHG emissions. The bottom-up methods include the mass balance method and emission factor method recommended by the IPCC National Greenhouse Gas Inventory Guidelines and are used based on the acquired chemical substance sales data or market data. The top-down methods include the tracer ratio correlation method, model inversion method, and box model method and are used based on the observed concentration data and the atmospheric behaviour of substances. Both types of emission calculation methods have advantages and disadvantages. Therefore, the 2019 Refinement to the 2006 IPCC Guidelines for National Greenhouse Gas Inventories proposed for the first time a method for retrieving greenhouse gas emissions based on atmospheric concentration to verify the bottom-up inventory results.

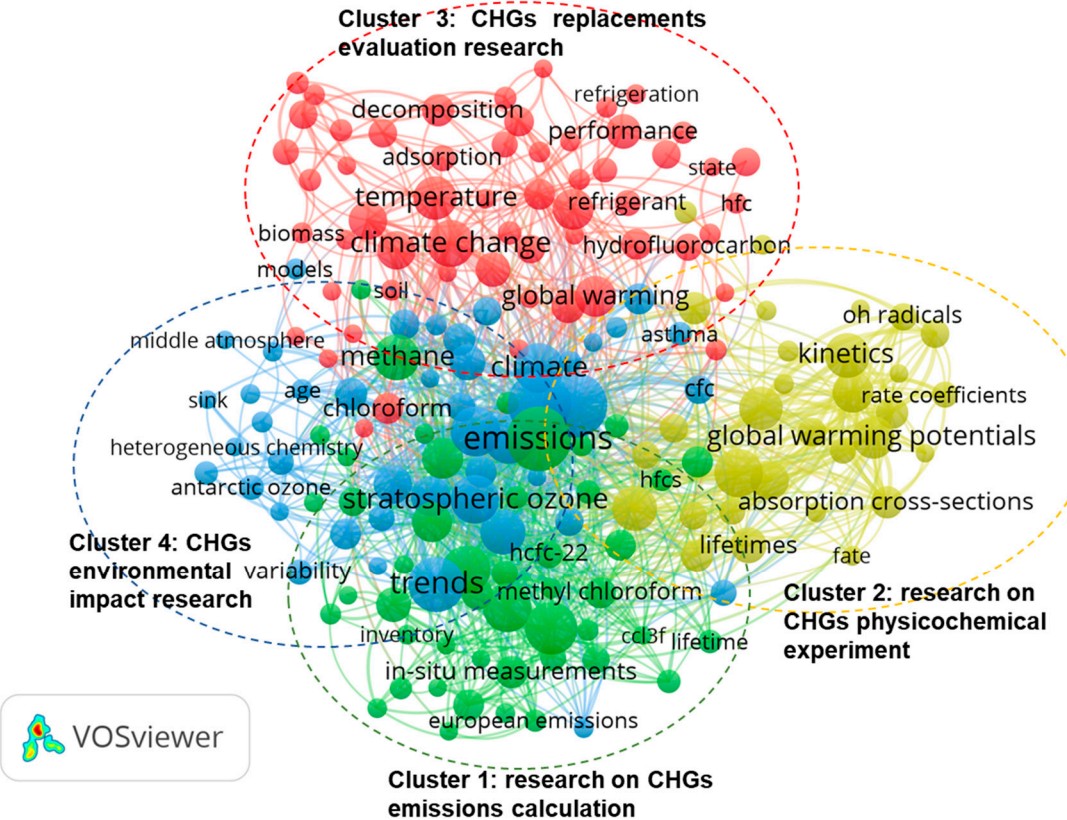

**Figure 6.** Cluster of CHGs co-occurrence keywords. The sizes of the nodes in Figure 6 represent the weights of the nodes. The larger the node, the larger the weight. The line between two keywords indicates that they appeared together. The thicker the line, the more co-occurrence they have.

Researchers often select appropriate emission calculation methods based on different research scales. The emission factor method and tracer ratio correlation method are mostly used to estimate emissions at the city or country scale [32–38]; the model inversion method is suitable for national and regional emission estimation [38–44]; and the box model is suitable for the estimation of emissions at the global scale.

### 4.2.2. Cluster 2 (Yellow): Physicochemical Analysis of CHGs

Keywords: atmospheric chemistry, kinetics, global warming potential, lifetime, rate constants, degradation, gas-phase reaction.

The research objective of physical and chemical experiments is to obtain the reaction rate, atmospheric lifespan, and radiation efficiency of the CHGs. These are important parameters for determining the global warming potential (GWP) and ozone-depleting substance potential (ODP) of CHGs. Some scholars have also used physical and chemical experiments to study the technologies for the removal of CHGs. In the study of reaction rate and atmospheric life, chemical experimental methods such as the relative rate method and absolute rate method are often used to determine the reaction rate of CHGs with OH radicals or chlorine atoms [45–53]. The reaction rate determines the atmospheric lifetime of a substance. In terms of radiation intensity efficiency, physical experimental methods, such as Fourier infrared spectroscopy, are often used to measure the infrared absorption cross-section of CHGs. In cases where the infrared (IR) spectrum cannot be measured, the radiation efficiency (RE) estimation based on computational chemistry is also very effective. In the study of halide molecule removal, chemical techniques such as thermal combustion or incineration are the most widely used [54–56]. These reactions often produce large quantities of complex chlorinated products; therefore, researchers are exploring combined processes or new alternatives, such as plasma-assisted technologies [57–60]. In addition,

catalytic Decomposition, photooxidation, and biodegradation have also been studied as removal technologies [61–63].

### 4.2.3. Cluster 3 (Red): Evaluation of CHG Replacements

Keywords: global warming, climate change, refrigerant, temperature, system, performance, energy, dynamics, simulation.

Halogenated gases are widely used in refrigeration, foaming, and fumigation. However, due to its environmental impacts, such as ozone depletion and global warming, the search for replacements with little or no environmental impacts has gradually become a research hotspot. Performance tests are used to evaluate the feasibility of a substance as a replacement for halogenated gases. Refrigerant and synonyms appeared 41 times, higher than the words for other consumer applications, indicating that research on the replacement of refrigerants is the most popular research topic in this regard. The indicators that need to be examined for refrigerants replacement includes the cooling capacity, coefficient of performance, consumption, volumetric efficiency of the compressor, and safety [64–72]. For example, the latest experimental results show that R744, HFO-1234yf, etc. can replace the existing refrigerant in the use of refrigeration equipment. Research on the replacement of blowing agents is also a hot spot, which can be seen from the fact that blowing agents and their synonyms have appeared for 20 times. The substitution effect of blowing agents is usually evaluated from the aspects of foam opening rate, thermal insulation, and foam size stability [73]. For example, azeotropic mixtures such as HCFC-142b and HCFC-22 can be used instead of CFC-12 in the foam. The replacements for fumigants were evaluated in terms of soil fungal population, microbial biomass C (MBC), respiration, nitrification potential, and changes in enzyme activity after using various fumigants.

### 4.2.4. Cluster 4 (Blue): Environmental Impact Research on CHGs

Keywords: chlorofluorocarbons, ultraviolet radiation, stratospheric ozone, trends, recovery, destruction, toxic, asthma.

The harm caused by increased UV radiation, impact of climate change, and biological toxicity of CHGs are the main research directions in the field of environmental impacts of CHGs.

The depletion of stratospheric ozone caused by CHGs results in an increase in the UV radiation flux. Many scholars have studied the effects of increased UV radiation flux on human, plant, animal, and microbial growth [3,74–81]. The research on the impact of climate change mainly focuses on the surface temperature changes caused by greenhouse gas emissions, contribution of radiative forcing, and adverse impacts of global warming on microorganisms and plants [82–88]. The toxicology of halides has been well studied, and the methods of biotoxicity assessment often include exposure experiments on mice or follow-up studies on long-term exposures [89–95]. Studies have shown that human exposure to specific halides may increase the risk of immune-mediated hepatitis or lead to tumorigenesis and/or have severe toxic effects on reproductive systems.

## 5. Trends of Hotspots

Exploring research hotspots in different periods is very necessary to understand a re-search field. We have counted the numbers of each word appearing in title, author keywords, abstract, keywords plus in multiple consecutive time periods (1999–2003, 2004–2008, 2009–2013, 2014–2018), and then "Aggregate class" is identified. "Aggregate class" can represent a possible research hotspot, including important synonym words and phrases (supporting words), which is often summarized by the professional researchers in this field. Finally, an overview of research hotspots was revealed by analyzing the number of publications containing these supporting words [22]. In this study, two hot themes are obtained, including research substances and environmental impact.

### 5.1. Research Substances

The Aggregate class of Research substances consists of "chlorofluorocarbons", "CFCs", "CFC-11" and "CFC-12", "halons" and "CCl$_4$" and their substitutes, HCFCs, HFCs, and HFOs constitute the research hotspots of research substances. As shown in Figure 7 that most of the early studies focused on the first generation of ODS, and the research object of the first generation of ODS accounted for more than 50% of the literature from 1999 to 2004. With the control and elimination of the first generation of ODS by the Montreal Protocol, the proportion of literature with the first generation of ODS as the research object decreased gradually, accounting for 29% in 2018. In contrast, substitutes for HCFCs, HFCs, and HFOs have attracted more attention, and the number of articles has increased significantly. In 1999, the percentage of HCFCs literature was 10% and peaked in 2011, accounting for 21%. Compared with HFCs and HFOs, which are gradually being widely used, the proportion of research on HCFCs has decreased, with an average of 16% in 2012–2018. In 1999, HFCs literature accounted for 15% and HFOs literature accounted for 10%. By 2018, HFCs literature accounted for 35% and HFOs literature accounted for 21%.

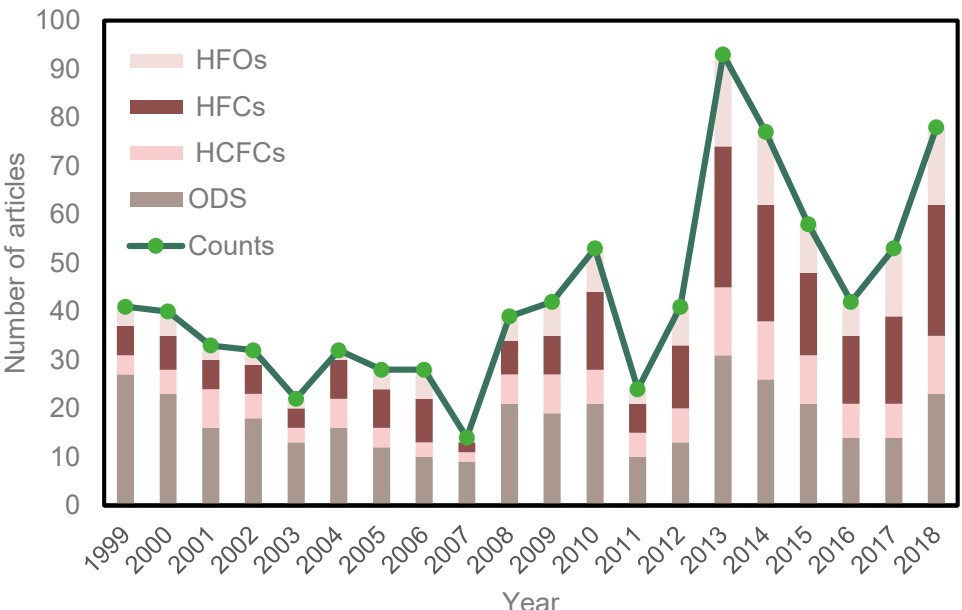

**Figure 7.** Number of articles published annually on different research substances.

### 5.2. Environmental Impact

The Aggregate class of Research substances consists of "ozone depletion", "global warming", "global warming potential" and "biological toxicity" and "inhalation toxicity" constitute the research hotspots of environmental impact. When the halogenated gases reach the stratosphere, they are exposed to ultraviolet radiation, and photolysis produces halogen radicals. Excessive halogen radicals accelerate the decomposition of ozone and destroy the balance between the generation and decomposition of the ozone layer. Furthermore, the absorption and reflection of infrared radiation by CHGs strengthened the greenhouse effect. CHGs are also biologically toxic. In this study, the annual literature on ozone depletion, global warming and biotoxicity was extracted to reveal hot research directions. Based on Figure 8, in 1999, the various environmental impacts of CHGs have been noticed. However, at this time, neither of them received much attention, with less than 20 papers published annually. As the impact of global warming became more apparent, research on the greenhouse effect of halides increased sharply, with the highest number of papers (58) being published in 2013. The number of publications on ozone depletion research and biological toxicity research is slowly increasing, and the trends of the two are very similar. Ozone depletion research reached its peak in 2014 (16 articles), and biological toxicity research reached its peak in 2013 (20 articles).

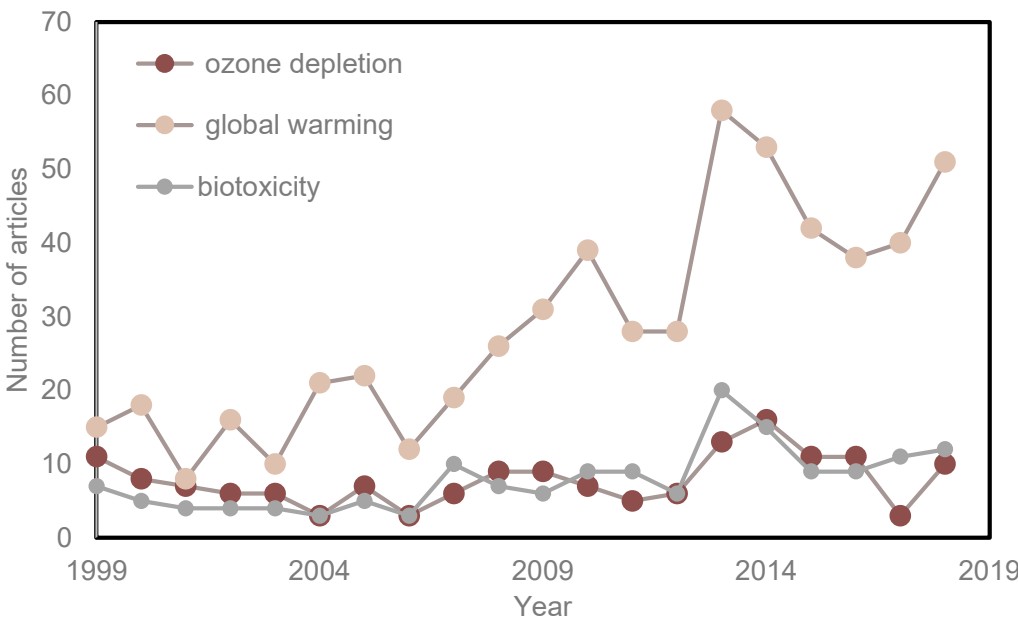

**Figure 8.** Number of articles published annually on different environmental impacts.

## 6. Conclusions

Based on 734 SCI publications on CHGs, this study provides an overview of the research on CHGs using two bibliometrics software VOSviewer and Sci2. The results showed that the United States was the most productive country, followed by UK and China. China has shown a strong growth momentum over the past decade. NOAA had the largest number of publications, followed by MIT and the University of Bristol. The most prolific authors were McCulloch, A, Ray F. Weiss, O'Doherty, S. and the authors who collaborated more with others mainly focused on the atmospheric concentrations and emissions of CHGs.

Using cluster analysis of all keywords and reading the articles in each cluster, four research hotspots of CHGs were identified and reviewed: (1) emissions calculation, (2) physicochemical analysis of halocarbons, (3) evaluation of replacements, and (4) environmental impact. Two types of methods are often used to estimate CHG emissions: top-down and bottom-up methods, and they can be used to verify each other's results. Physicochemical experiments are mainly carried out to study the removal process of CHGs and to obtain the key parameters, including reaction constants, atmospheric lifetime, and radiation efficiency, which can be used to calculate GWP and ODP. The purpose of replacement research is to find suitable substances to replace CHGs. The main evaluation index is the appraised index that includes cooling capacity, coefficient of performance, consumption, volumetric efficiency of the compressor, and safety. The environmental impact of CHGs is generally focused on ozone depletion, global warming, and biological toxicity.

The emerging topics and changes in research trends are closely related to the phase-out schedule of the Montreal Protocol. The original research topic was mainly CFCs and other ozone-depleting substances. With the phase-out of CFCs, research of HCFCs, as the transitional substitutes of ODSs, has gradually increased. Around the year when HCFCs began to freeze under the Montreal Protocol (2009 and 2010), research on HCFCs was gradually replaced by that on HFCs, HFOs, and other new substances. Global warming has always been the most concerning research hotspot, while research on ozone depletion shows a gradually rising trend. A total of 189 journals published articles on CHGs, referring to 90 disciplines, and the main disciplines were environmental science and physical science.

The purpose of this study is to provide an analysis of the publication knowledge related to CHGs. In addition, it provides guidance for researchers who want a comprehensive and quick understanding of the field.

**Author Contributions:** J.W. (Jing Wang): Conceptualization, Methodology, Investigation, Data curation, Writing-original draft, Validation. H.-Z.F.: Conceptualization, Methodology, Investigation, Data curation, Writing-original draft, Validation. J.X.: Investigation, Data curation, Writing-original draft. D.W.: Investigation, Data curation, Writing—original draft. Y.Y.: Investigation, Data curation, Writing-original draft. X.Z.: Supervision, Project administration. J.W. (Jing Wu): Conceptualization, Methodology, Supervision, Writing—reviewing and editing, Project administration, Validation. All authors have read and agreed to the published version of the manuscript.

**Funding:** This work was supported by the National Key R&D Program of China (Grant No. 2019YFC0214500), the National Natural Science Foundation of China (NSFC) (Grant No. 21976053 and No. 71804163), and Fundamental Research Funds for the Central Universities (Grant No. 2019MS042).

**Institutional Review Board Statement:** Not applicable.

**Informed Consent Statement:** Not applicable.

**Data Availability Statement:** The datasets analysed during the current study are available in the Science Citation Index (SCI) Expanded database of the Web of Science, https://www.webofscience.com/wos/woscc/basic-search (accessed on 24 September 2019).

**Conflicts of Interest:** The authors declare no conflict of interest.

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
