# Peer review of "Trends of Studies on Controlled Halogenated Gases under International Conventions during 1999–2018 Using Bibliometric Analysis: A Global Perspective"

_sustainability, doi:10.3390/su14020806_

Round 1

Reviewer 1 Report

The authors picked 2 bibliometric analysis tools to study the recent publications on controlled halogenated gases, from the perspectives of ozone depletion and global warming. The used methodology is simple, but the content and message are interesting and important. The paper is written in a clear way. So, I support this paper after the authors enlarge the font sizes of Figure 3-6.

Reviewer 2 Report

In this paper, the authors searched several keywords related to the halogenated gases in the SCIE database of WoS and got 734 publication results after the selection. Then the bibliometric analysis was done to describe the collaborations of the countries, institutions, authors and keywords. Based on the expert knowledge, the keywords network was classified into 4 clusters. The author introduced the search contents of the 4 clusters and the research trends of 2 of them. The paper concentrates on the title and gives the readers the general research changed trend of the halogenated gases from 1999 to 2018. However, the manuscript, in its present form, contains several weaknesses. Appropriate revisions to the following points should be undertaken in order to justify recommendation for publication.

  1. The research trend analysis seems depends on the subjective judgment (line 341, 383-386). More data should be presented to support the opinions.
  2. The research period is old. Please add the newest data into the analysis.
  3. The figures could not give the readers the concise and highlight information. They should be modified, such as figure 2, 3, 4, 5, 6.
  4. The authors names in section 3.3 should be uniformed. Some of them can’t be found in the figure 4. The line 249-253 only describe two clusters. How about the other clusters?
  5. The line 279-283 should be the caption of figure 6.
  6. What’s the meaning of line 387-392?
  7. The downward researches trend about the HCFCs was support by the decrease proportion of the publications. But the publication number seems increase in terms of figure 7. Therefore, I don’t think the trend is decrease.
  8. The research trend of only focus on two clusters. How about the rest two clusters?

Reviewer 3 Report

This study provides a nice and important overview of the available research with regards to halogenated gases, an important type of gases that cause ozone depletion and contribute to global warming. I would suggest only some remarks for improving the manuscript. Thank you.

1. It is apparent from Figure 2 that less research is available at emerging countries e.g. in the South East Asia. The region is having rapid development and industrialization which means that substantial halogenated gases can potentially being released from that region and cause harm to the ozone layer. From the analysis, is there any indication onto why research is less developed in that region and what are authors opinion/recommendation to resolve the issue. It would be interesting if more research in that region/other regions can be developed further and international cooperation can be fostered with more established research team e.g. in UK and US.
2. Line 12 : To remove ‘delete’ and kindly revise the sentence accordingly.
3. Line 27 : I think "ozone-depleting substances" would be more appropriate than " ozone depletion".
4. Line 37 : Perhaps “the ozone concentrations” is more suitable than “the ozone content”?
5. Line 47: To define CCl4 .
6. Line 60: To replace “the” with “The”.
7. Line 62 and 63 : “However, recent scientific studies have found that some CHGs concentration and emission trends are different from those expected”. To briefly explain why is that so?
8. Line 213 : To provide the full name of “Natl Ctr Atmospher Res”.
9. Line 236 – 237: Please provide legend for the Figure 3.
10. Line 255-256: Please provide legend for the Figure 4.
11. Line 381: To replace “ozonation-depleting substances” with “ozone-depleting substances”.

Reviewer 4 Report

Comments on manuscript “Trends of halogenated gases under international conventions during 1999-2018 using bibliometric analysis: A global perspective” (sustainability-1479050)

This manuscript provides an overview on the trends of CHGs. Its contents are suitable with the journal’s scope. I have some specific comments below.

Title: In my opinion, the title does not reflect the manuscript content properly. Please consider to revise “Trends of studies on controlled halogenated gases …”. With the current title, I expect to obtain information on the contamination status and distribution trends of CHGs in the global environment. However, this study mainly summarized general information about the studies but not research contents of such studies from an environmental science perspective.

Abstract: The abstract should be revised. It seems to be a summary of review strategy and methodology. Please provide major findings and evaluation sentences.

Part 5, Trends in research content: Besides “research substances” and “environmental impact”, if possible, please describe about other aspects such as study locations, spatiotemporal trends of CHG contamination, etc.

Figure 8: If possible, please add information on “biological toxicity”, as described in Line 443-444.
